# Assessing the Robustness of a Factory Amid the COVID-19 Pandemic: A Fuzzy Collaborative Intelligence Approach

**DOI:** 10.3390/healthcare8040481

**Published:** 2020-11-12

**Authors:** Toly Chen, Yu-Cheng Wang, Min-Chi Chiu

**Affiliations:** 1Department of Industrial Engineering and Management, National Chiao Tung University, 1001, University Road, Hsinchu 30010, Taiwan; tolychen@ms37.hinet.net; 2Department of Aeronautical Engineering, Chaoyang University of Technology, Taichung 413310, Taiwan; 3Department of Industrial Engineering and Management, National Chin-Yi University of Technology, Taichung 41170, Taiwan; mcchiu@ncut.edu.tw

**Keywords:** COVID-19 pandemic, robustness, fuzzy collaborative intelligence, wafer fabrication

## Abstract

The COVID-19 pandemic has affected the operations of factories worldwide. However, the impact of the COVID-19 pandemic on different factories is not the same. In other words, the robustness of factories to the COVID-19 pandemic varies. To explore this topic, this study proposes a fuzzy collaborative intelligence approach to assess the robustness of a factory to the COVID-19 pandemic. In the proposed methodology, first, a number of experts apply a fuzzy collaborative intelligence approach to jointly evaluate the relative priorities of factors that affect the robustness of a factory to the COVID-19 pandemic. Subsequently, based on the evaluated relative priorities, a fuzzy weighted average method is applied to assess the robustness of a factory to the COVID-19 pandemic. The assessment result can be compared with that of another factory using a fuzzy technique for order preference by similarity to ideal solution. The proposed methodology has been applied to assess the robustness of a wafer fabrication factory in Taiwan to the COVID-19 pandemic.

## 1. Introduction

The COVID-19 pandemic has severely affected many industries, such as tourism, aviation, telemedicine, and catering [1,2,3,4,5], and manufacturing was no exception. Factories all over the world were, to varying degrees, affected by the COVID-19 pandemic. Among them, many factories were forced to shut down to avoid the spread of COVID-19 [6]. If a closed factory is located in the upstream or midstream of a supply chain, then downstream factories will also be affected [7], ending up with troubles in delivering final products to customers [8,9]. As a result, the impact of the COVID-19 pandemic on a supply chain is long term, uncertain, and will propagate disruptively [10]. In addition, some factories have transformed their production lines to make personal protection equipment or medical equipment, such as facial masks and medical ventilators, to mitigate such effects [11]. In the view of Malik et al. [12], it is easier for factories equipped with collaborative robots (or cobots) to perform such conversions. Further, notebook factories have even benefited from the increased demand for distance teaching/learning and conferences [13].

Obviously, the impact of the COVID-19 pandemic on different types of factories was not the same. Some factories were more vulnerable to the COVID-19 pandemic [14], while others even benefited in the short term [15]. For example, labor-intensive industries are vulnerable to the COVID-19 pandemic due to frequent contact between workers [14]. On the contrary, automated factories are robust to the COVID-19 pandemic [12]. In addition, factories with workers that are economically disadvantaged will avoid taking time off in the early stages of infection, which will increase the risk of large-scale infections [16]. Therefore, factories full of such workers are also vulnerable to the COVID-19 pandemic. A factory will also be vulnerable if the demand for its products decreases amid the COVID-19 pandemic. In addition, factories that rely on a large number of migrant workers are vulnerable because these workers came from various sources of infection [17]. Nevertheless, recently, great progress has been made in the development of COVID-19 vaccines [18], which will greatly reduce the vulnerability of factories. In sum, although some short-term measures can be taken, such as wearing masks and physical distancing, to temporarily strengthen the robustness of a factory to the COVID-19 pandemic, there are some basic and systemic factors in determining the factory’s robustness. In order to assess the robustness of a factory to the COVID-19 pandemic, a fuzzy collaborative intelligence approach is proposed in this study. Here, the robustness of a factory to the COVID-19 pandemic is defined as “the degree to which the normal operation of a factory is not affected by the COVID-19 pandemic without taking short-term preventive measures”. This topic has rarely been investigated in the past. Similar issues have only been discussed at the government level [19]. This topic is importance because
(1)Factories that are not robust to the COVID-19 pandemic suffered workforce losses, shortages of raw materials and even factory closures. Therefore, a factory should strive to enhance its robustness to the COVID-19 pandemic. For this, assessing the robustness of the factory is a prerequisite;(2)Factories should cooperate with customers and suppliers who are robust to the COVID-19 pandemic to reduce operational risks. Therefore, it is also important to assess the robustness of a customer (or supplier) to the COVID-19 pandemic.

The motives for proposing the fuzzy collaborative intelligence approach are explained as follows. First, the robustness of a factory to the COVID-19 pandemic is obviously a subjective concept affected by many quantitative and qualitative factors. In addition, the robustness is also subject to unpredictable changes inside and outside the factory [10,20]. For these reasons, using a fuzzy set instead of a probabilistic set to model the robustness of a factory to the COVID-19 pandemic is an appropriate treatment. In the literature, Zhang et al. [21] and Yu et al. [22] used hesitant fuzzy numbers to represent subjective assessments. In addition, the linguistic sets used by experts might vary. The adoption of more complex forms of fuzzy numbers can theoretically elevate the flexibility of the decision-making process, which is based on the premise that decision makers can understand and accept such fuzzy numbers [20]. In addition, multiple experts’ opinions are consulted to avoid personal bias or omissions [23], resulting in the collaborative nature of the proposed methodology [24,25]. Further, fuzzy collaborative intelligence methods have been successfully applied to various topics, e.g., retrieving concepts from medical documents [26], enhancing the performance of a ubiquitous location-aware service system [27], distributing tasks among robots or three-dimensional (3D) printers [28,29,30,31], evaluating the suitability of a smart technology application for fall detection [32], etc.

In the proposed fuzzy collaborative intelligence approach, each expert applies the fuzzy geometric mean (FGM) method [33] to evaluate the relative priorities of factors that affect the robustness of a factory to the COVID-19 pandemic. If all experts have reached an overall consensus, fuzzy intersection (FI) [34] is applied to aggregate their evaluation results. Otherwise, the partial-consensus FI (PCFI) approach [35,36] will be applied to achieve the same goal. Subsequently, based on the aggregation result, the fuzzy weighted average (FWA) method is applied to assess the robustness of a factory to the COVID-19 pandemic. The robustness of a factory can also be compared with that of another using the fuzzy technique for order preference by similarity to ideal solution (FTOPSIS) [37,38]. Specifically speaking, a prevalent assessment technique (i.e., FTOPSIS) is applied to assess the robustness of a factory to the COVID-19, for which a fuzzy collaborative intelligence approach is proposed to provide the weights of factors in the ordinary assessment technique. The combination of FGM (i.e., fuzzy analytic hierarchy process, FAHP) and FWA (or fuzzy TOPSIS) has been widely applied to various assessment tasks in many fields, such as supplier assessment [39], thermal power plant location assessment [40], plastics recycling method assessment [41], etc.

The assessment of a factory’s robustness to the COVID-19 pandemic is based on the following assumptions (or scenarios):(1)The COVID-19 pandemic in the region the factory is located will not worsen significantly compared to the past.(2)The government in the region can contain the COVID-19 pandemic as in the past.(3)In the foreseeable future, the availability of vaccines for COVID-19 can be expected.

Over the existing methods, the proposed methodology has the following advantages:(1)Most existing FAHP methods aggregate the pairwise comparison results by experts without checking whether there is a consensus, which may lead to an unacceptable decision. On the contrary, in the proposed fuzzy collaborative intelligence approach, FI is applied to find the consensus among experts before aggregation. In this way, the final decision-making result will be acceptable to all experts.(2)In the existing FTOPSIS methods, the order of the objective function is determined subjectively, which considerably affects the decision-making result. On the contrary, in the proposed methodology, the order of the objective function is varied to observe the change of the decision-making result, thereby generating a more reliable decision.

One disadvantage of the proposed methodology is that the computation complexity is slightly increased.

The remainder of this paper is organized as follows. Section 2 is devoted to discussing factors that influence the robustness of a factory to the COVID-19 pandemic. Section 3 introduces the fuzzy collaborative intelligence approach for assessing the robustness of a factory to the COVID-19 pandemic. Section 4 reports the details of applying the proposed fuzzy collaborative intelligence approach to a real case to illustrate its applicability. Section 5 concludes this study and puts forth some possible topics for future investigation.

## 2. Factors Influencing the Robustness of a Factory to the COVID-19 Pandemic

Factors influencing the robustness of a factory to the COVID-19 pandemic can be classified into four categories (see Figure 1):
(1)Demand shrinkage: The demand for certain types of products diminished after the outbreak of the COVID-19 pandemic. For example, reduced cross-country travel led to lower demand for transportation fuels, airplanes, and cruises [42,43]. In addition, people’s demand for luxury goods also decreased, and related factories were forced to shut down or transform [44]. Further, the demand for personal computers (PC) was replaced by that for notebooks and pads for distance teaching/learning and conferencing [45].(2)Worker health risks: An infected worker caused the workers who come in contact with him/her to be quarantined [46], which leads to a great loss of factory capacity. If many workers are infected, a factory may have to be closed [47]. In order to solve this problem, wearing masks and physical distancing are short-term measures that can be taken. However, in labor-intensive factories, physical distancing is a challenging task. In addition, Singapore’s experience has shown that migrant workers living in dormitories face additional health risks [48], because they may be from countries with the severe COVID-19 pandemic or be infected while taking airplanes or living in dormitories. Most short-term preventive measures and intervention strategies, such as wearing masks, physical distancing, periodic disinfection, and regular temperature checks, are designed to reduce such risks [49].(3)Supply chain breakage: The COVID-19 pandemic has caused the closures of factories around the world, many of which are located in the upstream or midstream segments of supply chains [50]. As a result, the supply cost of downstream factories has increased to compete for the limited capacity [51]. Ivanov [10] conducted a simulation study to evaluate and predict the impact of an epidemic outbreak on a global supply chain. In order to mitigate the impact, finding alternative suppliers has become a critical task. Some supply chain researchers have listed actions that supply chain participants can take in the short and long terms [7,51], as summarized in Table 1.(4)Factory shutdown: In the COVID-19 pandemic, there are basically four types of factory shutdown (see Figure 2). First, many factories were closed because workers were tested positive [47,52]. In this regard, factories that handle (fresh) food seemed to be particularly vulnerable to the COVID-19 pandemic [47,53]. Second, due to the lack of raw materials amid the COVID-19 pandemic, some factories have closed [54]. These factories were undoubtedly located in the midstream or downstream segments of supply chains. Third, many regional governments have forced factories to close to check whether their working environments are prone to spreading COVID-19 or disinfect these factories [55]. Fourth, during the COVID-19 pandemic, the demand for certain products (such as fashion brands and PCs) has decreased [56]. Factories manufacturing such products can only be closed.

## 3. The Fuzzy Collaborative Intelligence Approach

The fuzzy collaborative intelligence approach used to assess the robustness of a factory to the COVID-19 pandemic comprises three main parts: the FGM method for evaluating the relative priorities of factors that affect the robustness of a factory to the COVID-19, the FI (or PCFI) method for aggregating the relative priorities evaluated by experts, and the FWA (or FTOPSIS) method for assessing (or comparing) the robustness of a factory to the COVID-19 pandemic, as shown in Figure 3. The three parts are described in the following subsections.

The steps for implementing the proposed methodology are as follows:Step 1.(Each expert) Evaluate the relative priorities of critical factors using FGM.Step 2.If the evaluation results are consistent, go to Step 3; otherwise, return to Step 1.Step 3.If all experts reach an overall consensus, apply FI to aggregate experts’ evaluation results; otherwise, apply PCFI to aggregate experts’ evaluation results.Step 4.Apply FWI to assess the robustness of the target factory to the COVID-19 pandemic.Step 5.Apply FTOPSIS to compare the robustness of the target factory to that of another factory.

### 3.1. FGM for Evaluating the Relative Priorities of Factors

In the fuzzy collaborative intelligence approach, multiple experts assess the robustness of a factory to the COVID-19 pandemic from their own points of view. At first, each experts evaluates the relative priorities of factors affecting the robustness of a factory to the COVID-19 pandemic in pairs using the FGM method. Such evaluations are given in linguistic terms such as “as equal as,” “weakly more important than,” “strongly more important than,” “very strongly more important than,” “absolutely more important than,” etc. A common way is mapping these linguistic terms to triangular fuzzy numbers (TFNs) within [1,9,57]:(1)$${\tilde{a}}_{ji}=(a_{ij1},\;a_{ij2},\;a_{ij3})$$
${\tilde{a}}_{ji}=(a_{ij1},\;a_{ij2},\;a_{ij3})$ is the relative priority of factor *i* over factor *j*. The symbol “~” indicates a fuzzy variable. In addition,
(2)$$0\leq a_{ji2}\leq 9$$
(3)$$a_{ij1}=\{\begin{array}{ccc} 1/a_{ji3} & if & a_{ij2}<1\\ \max(a_{ij2}-2,\;1) & otherwise & \end{array}$$
(4)$$a_{ij3}=\{\begin{array}{ccc} 1/a_{ji1} & if & a_{ij2}<1\\ \min(a_{ij2}+2,\;9) & otherwise & \end{array}$$

These TFNs can be expanded to increase the possibility for experts to reach a consensus [58]. In contrast, Samanlioglu and Kaya [59] narrowed the ranges of these TFNs to improve the consistency of pairwise comparison results.

Based on pairwise comparison results, a fuzzy judgment matrix ${\tilde{A}}_{n\times n}=[{\tilde{a}}_{ij}]$ is constructed. The fuzzy eigenvalue and eigenvector of $\tilde{A}$, indicated with $\tilde{\lambda }$ and $\tilde{x}$, respectively, satisfy
(5)$$det(\tilde{A}(-)\tilde{\lambda }I)=0$$
and
(6)$$(\tilde{A}(-)\tilde{\lambda }I)(\times )\tilde{x}=0$$
where (−) and (×) denote fuzzy subtraction and multiplication [60], respectively: (7)$$\tilde{B}(-)\tilde{C}=(B_{1}-C_{3},\;B_{2}-C_{2},\;B_{3}-C_{1})$$
(8)$$\tilde{B}(\times )\tilde{C}=(B_{1}C_{1},\;B_{2}C_{2},\;B_{3}C_{3})\;\mathrm{if}\;B_{1},\;C_{1}\geq 0$$
where $\tilde{B}$ and $\tilde{C}$ are two TFNs. 

The FGM method [33] is applied to derive the relative priority of each factor (${\tilde{w}}_{i}$) as
(9)$${\tilde{w}}_{i}=\frac{\sqrt[{n}]{\prod_{j=1}^{n}{\tilde{a}}_{ij}}}{\sum_{k=1}^{n}\sqrt[{n}]{\prod_{j=1}^{n}{\tilde{a}}_{kj}}}$$

According to the arithmetic for TFNs, the higher-order root of a TFN $\tilde{B}$ can be calculated as [60]
(10)$$\sqrt[{n}]{\tilde{B}}=(\sqrt[{n}]{B_{1}},\;\sqrt[{n}]{B_{2}},\;\sqrt[{n}]{B_{3}})$$

Therefore, Equation (9) can be decomposed into [61]
(11)$$w_{i1}=\frac{1}{1+\sum_{k\neq i}\frac{\sqrt[{n}]{\prod_{j=1}^{n}a_{kj3}}}{\sqrt[{n}]{\prod_{j=1}^{n}a_{ij1}}}}$$
(12)$$w_{i2}=\frac{1}{1+\sum_{k\neq i}\frac{\sqrt[{n}]{\prod_{j=1}^{n}a_{kj2}}}{\sqrt[{n}]{\prod_{j=1}^{n}a_{ij2}}}}$$
(13)$$w_{i3}=\frac{1}{1+\sum_{k\neq i}\frac{\sqrt[{n}]{\prod_{j=1}^{n}a_{kj1}}}{\sqrt[{n}]{\prod_{j=1}^{n}a_{ij3}}}}$$

In addition, fuzzy maximal eigenvalue ${\tilde{\lambda }}_{\max}$ (i.e., the maximum possible value of $\tilde{\lambda }$) can be derived as [61]
(14)$${\tilde{\lambda }}_{\max}=\frac{1}{n}\sum_{i=1}^{n}\frac{\sum_{j=1}^{n}\left({\tilde{a}}_{ij}(\times ){\tilde{w}}_{j}\right)}{{\tilde{w}}_{i}}$$
which is equivalent to
(15)$$\lambda_{\max,1}=1+\frac{1}{n}\sum_{i=1}^{n}\sum_{j\neq i}\frac{a_{ij1}w_{j1}}{w_{i3}}$$
(16)$$\lambda_{\max,2}=1+\frac{1}{n}\sum_{i=1}^{n}\sum_{j\neq i}\frac{a_{ij2}w_{j2}}{w_{i2}}$$
(17)$$\lambda_{\max,3}=1+\frac{1}{n}\sum_{i=1}^{n}\sum_{j\neq i}\frac{a_{ij3}w_{j3}}{w_{i1}}$$

Pairwise comparison results are consistent if ${\tilde{a}}_{ij}(\times ){\tilde{a}}_{jk}={\tilde{a}}_{ik}$ ∀ *i*, *j*, *k*. The consistency of pairwise comparison results can be evaluated in terms of consistency ratio (CR) as [62]
(18)$$\tilde{CR}=\frac{\frac{{\tilde{\lambda }}_{\max}(-)n}{n-1}}{RI}$$
where *RI* is random consistency index [62]. $\tilde{CR}$ is a positive TFN. However, $\tilde{CR}$ may be negative if an approximation technique, such as FGM, is applied to estimate the value of ${\tilde{\lambda }}_{\max}$. This problem can be solved by applying exhaustive computation techniques (such as alpha-cut operations) [63]. For pairwise comparison results to be consistent, $\tilde{CR}$ should be less than 0.1~0.3, depending on the problem size and uncertainty [64].

### 3.2. FI (or PCFI) for Aggregating the Relative Priorities Evaluated by Experts

If the (overall) consensus among experts exists, FI can be applied to aggregate the relative priorities evaluated by them as follows [34].

**Definition** **1.**
*The fuzzy intersection (FI) of the relative priorities evaluated by M experts for the i-th factor, indicated with*
${\tilde{w}}_{i}(1)$
*~*
w˜i(M)
*, is denoted by*
$\tilde{FI}(\{{\tilde{w}}_{i}(m)\})$
*such that*
(19)$$\mu_{\tilde{FI}(\{{\tilde{w}}_{i}(m)\})}(x)=\underset{m}{\min}\mu_{{\tilde{w}}_{i}(m)}(x)$$
*where*
$\mu_{\tilde{FI}(\{{\tilde{w}}_{i}(m)\})}(x)$
*and*
$\mu_{{\tilde{w}}_{i}(m)}(x)$
*denote the memberships of x in*
$\tilde{FI}(\{{\tilde{w}}_{i}(m)\})$
*and*
${\tilde{w}}_{i}(m)$
*, respectively.*


Otherwise, PCFI is applied to aggregate the relative priorities evaluated by most experts as follows [35].

**Definition** **2.**
*The H/M partial consensus fuzzy intersection (PCFI) of the relative priorities evaluated by M experts for the i-th factor, indicated with*
${\tilde{w}}_{i}(1)$
*~*
w˜i(M)
*, is denoted by*
${\tilde{PCFI}}^{H/M}(\{{\tilde{w}}_{i}(m)\})$
*such that*
(20)$$\mu_{{\tilde{PCFI}}^{H/M}(\{{\tilde{w}}_{i}(m)\})}(x)=\underset{all\;g}{\max}(\min(\mu_{{\tilde{w}}_{i}(g(1))}(x),\;...,\;\mu_{{\tilde{w}}_{i}(g(H))}(x)))$$
*where*
$\mu_{{\tilde{PCFI}}^{H/M}(\{{\tilde{w}}_{i}(m)\})}(x)$
*and*
$\mu_{{\tilde{w}}_{i}(g(h))}(x)$
*denote the memberships of x in*
${\tilde{PCFI}}^{H/M}(\{{\tilde{w}}_{i}(m)\})$
*and*
${\tilde{w}}_{i}(g(h))$
*, respectively; g() ∈ Z^+^; 1 ≤ g() ≤ M; g(p) ∩ g(q) = ∅ ∀ p ≠ q; H ≥ 2.*


The problem is how to determine the number of experts who have reached a partial consensus. Chen and Wu [36] believed that it is better to get more experts to reach a partial consensus, which is more difficult, and the PCFI result may only cover a few possible values. In order to cover a sufficient number of possible values, the range of the PCFI result should be greater than a threshold *ξ* [46].

An example is given in Figure 4, which shows the relative priorities evaluated by five experts. First, because the FI result is an empty set, there is a lack of consensus among all experts. In order to solve this problem, the partial consensus among any four experts is sought instead. The result is shown in Figure 5. The PCFI result is not empty. However, only values from 0.41 to 0.49, with a range of 0.08, are acceptable to experts that reach a partial consensus. Such a narrow range limits the flexibility of subsequent operations. To overcome this limitation, the threshold *ξ* is set to 0.20. Subsequently, the partial consensus among any three experts is also sought. The result is shown in Figure 6. The range of acceptable values extends from 0.08 to 0.33, which is wider than the threshold *ξ*.

However, $\tilde{PCFI}(\{{\tilde{w}}_{i}(m)\})$ is a polygonal fuzzy number, while ${\tilde{p}}_{qi}$ is a TFN. Their combination is not easy to calculate. To tackle such complexity, $\tilde{PCFI}(\{{\tilde{w}}_{i}(m)\})$ is approximated with a TFN as [49]:(21)$$\begin{array}{l} \tilde{PCFI}(\{{\tilde{w}}_{i}(m)\})\\ ≅\left(\begin{array}{l} \min(\tilde{PCFI}(\{{\tilde{w}}_{i}(m)\}),\;\\ 3COG(\tilde{PCFI}(\{{\tilde{w}}_{i}(m)\})-\max(\tilde{PCFI}(\{{\tilde{w}}_{i}(m)\})-\min(\tilde{PCFI}(\{{\tilde{w}}_{i}(m)\}),\;\\ \max(\tilde{PCFI}(\{{\tilde{w}}_{i}(m)\}) \end{array}\right) \end{array}$$
where $\min(\tilde{PCFI}(\{{\tilde{w}}_{i}(m)\})$, $COG(\tilde{PCFI}(\{{\tilde{w}}_{i}(m)\})$, and $\max(\tilde{PCFI}(\{{\tilde{w}}_{i}(m)\})$ denote the minimum, center of gravity (COG) [65] and maximum of $\tilde{PCFI}(\{{\tilde{w}}_{i}(m)\})$, respectively:(22)$$COG(\tilde{PCFI}(\{{\tilde{w}}_{i}(m)\}))=\frac{\int_{all\;x}x\mu_{\tilde{PCFI}(\{{\tilde{w}}_{i}(m)\})}(x)dx}{\int_{all\;x}\mu_{\tilde{PCFI}(\{{\tilde{w}}_{i}(m)\})}(x)dx}$$
as illustrated in Figure 7. In this way, the defuzzified value of the approximating TFN is equal to $COG(\tilde{PCFI}(\{{\tilde{w}}_{i}(m)\})$.

### 3.3. FWA (or FTOPSIS) for Assessing (and Comparing) the Robustness of A Factory Amid the COVID-19 Pandemic

Subsequently, the FWA method [39] is proposed to assess the robustness of a factory amid the COVID-19 pandemic:(23)$$\tilde{O}=\sum_{i=1}^{n}\left({\tilde{PCFI}}^{H/M}(\{{\tilde{w}}_{i}(m)\})(\times ){\tilde{p}}_{i}\right)$$
where $\tilde{O}$ is the overall performance of the factory; ${\tilde{p}}_{i}$ is the performance of the factory in optimizing the *i*-th factor. The COG method [65] can be applied to defuzzify $\tilde{O}$.

If there are several factories to be compared, then the FTOPSIS approach can be applied. First, the performance of each factory in optimizing a factor is normalized using the fuzzy distributive normalization [37]:(24)$$\begin{array}{c} {\tilde{\rho }}_{qi}=\frac{{\tilde{p}}_{qi}}{\sqrt{\sum_{\phi =1}^{Q}{\tilde{p}}_{\phi i}^{2}}}\\ =\frac{1}{\sqrt{1+\sum_{\phi \neq q}(\frac{{\tilde{p}}_{\phi i}}{{\tilde{p}}_{qi}}{)}^{2}}} \end{array}$$

According to the fuzzy arithmetic for TFNs,
(25)$$\rho_{qi1}=\frac{1}{\sqrt{1+\sum_{\phi \neq q}(\frac{p_{\phi i3}}{{\tilde{p}}_{qi1}}{)}^{2}}}$$
(26)$$\rho_{qi2}=\frac{1}{\sqrt{1+\sum_{\phi \neq q}(\frac{p_{\phi i2}}{{\tilde{p}}_{qi2}}{)}^{2}}}$$
(27)$$\rho_{qi3}=\frac{1}{\sqrt{1+\sum_{\phi \neq q}(\frac{p_{\phi i1}}{{\tilde{p}}_{qi3}}{)}^{2}}}$$
where ${\tilde{p}}_{qi}$ is the performance of the *q*-th factory in optimizing the *i*-th factor; ${\tilde{\rho }}_{qi}$ is the normalized performance. Subsequently, the fuzzy weighted score is calculated based on the relative priorities derived using the FI (or PCFI) approach:(28)$${\tilde{s}}_{qi}=\tilde{FI}(\{{\tilde{w}}_{i}(m)\})(\times ){\tilde{\rho }}_{qi}$$
or
(29)$${\tilde{s}}_{qi}=\tilde{PCFI}(\{{\tilde{w}}_{i}(m)\})(\times ){\tilde{\rho }}_{qi}$$

Subsequently, the fuzzy ideal (zenith) point and the fuzzy anti-ideal (nadir) point are specified, respectively, as [37,61]
(30)$${\tilde{\Lambda }}^{+}=\{{\tilde{\Lambda }}_{i}^{+}\}=\{\underset{q}{\max}{\tilde{s}}_{qi}\}$$
(31)$${\tilde{\Lambda }}^{-}=\{{\tilde{\Lambda }}_{i}^{-}\}=\{\underset{q}{\min}{\tilde{s}}_{qi}\}$$

The fuzzy distance from each factory to the two points are calculated, respectively, as [37,61]
(32)$${\tilde{d}}_{q}^{+}=\sqrt[{v}]{\sum_{i=1}^{n}\max{\left({\tilde{\Lambda }}_{i}^{+}(-){\tilde{s}}_{qi},\;0\right)}^{v}}$$
(33)$${\tilde{d}}_{q}^{-}=\sqrt[{v}]{\sum_{i=1}^{n}\max{\left({\tilde{s}}_{qi}(-){\tilde{\Lambda }}_{i}^{-},\;0\right)}^{v}}$$
where *v* ∈ *Z*^+^. In the traditional FTOPSIS method, *v* is set to 2. However, the value of *v* definitely affects the ranking result. By gradually increasing the value of *v*, the ranking result is expected to converge, as illustrated in Figure 8. When *v* is less than 5, the ranking result becomes different as the value of *v* changes. However, after the value of v is greater than 5, the ranking result converges and no longer changes.

Finally, the fuzzy closeness of each factory is obtained as [37,61]
(34)$${\tilde{C}}_{q}=\frac{{\tilde{d}}_{q}^{-}}{{\tilde{d}}_{q}^{+}(+){\tilde{d}}_{q}^{-}}$$

The fuzzy closeness is higher if the factory is farther from the fuzzy anti-ideal solution, but closer to the fuzzy ideal solution. A factory is more robust to the COVID-19 pandemic if its fuzzy closeness is higher. To get an absolute ranking, the fuzzy closeness can also be defuzzified using COG [65].

## 4. Application

The proposed methodology has been applied to assess the robustness of a wafer fabrication factory (wafer fab) located in Tainan Scientific Park, Taiwan, to the COVID-19 pandemic. The wafer fab belongs to one of the largest wafer foundries [66,67,68] in the world. The wafer foundry has twelve wafer fabs worldwide. The conditions of these wafer fabs are different and should be assessed separately. In the first quarter of 2020, the semiconductor industry has suffered considerable losses because of the severe COVID-19 pandemic in Chinese and Asian markets [65]. Whether the wafer fab is robust to the COVID-19 pandemic is still a question that needs to be evaluated urgently.

Basically, semiconductor products can be divided into six main types: networking and communication, data processing, industrial applications, consumer electronics, automotive, and government. The impact of the COVID-19 pandemic on these categories is different. In particular, the demand for consumer electronics and automotive products has fallen sharply during the COVID-19 pandemic, while the demand for networking and communication and data processing was less affected [69,70].

After reviewing the relevant literature and practices, the following factors have been considered to affect the robustness of a wafer fab to the COVID-19 pandemic:COVID-19 containment performance: the performance of the local government in containing the COVID-19 pandemic [19];Pandemic severity: the current severity of the COVID-19 pandemic in the region [71];Vaccine acquisition speed: the estimated acquisition speed of COVID-19 vaccines [72];Demand shrinkage: the extent to which the demand for major products has been affected by the COVID-19 pandemic [45,65];Supplier impact: the extent on raw material suppliers [7,51];Infection risk: mainly due to interactions with foreign visitors (customers or suppliers) and between employees (especially when there are migrant workers) [16,48].

Three experts compared the relative priorities of these factors with linguistic terms. The results are summarized in Table 2.

Based on pairwise comparison results, the following fuzzy judgment matrixes were constructed:$$\tilde{A}(1)=\left[\begin{array}{cccccc} 1 & 1/(1,\;3,\;5) & 1/(2,\;4,\;6) & \left(2,\;4,\;6\right) & \left(3,\;5,\;7\right) & \left(3,\;5,\;7\right)\\ \left(1,\;3,\;5\right) & 1 & 1/(3,\;5,\;7) & 1/(1,\;3,\;5) & \left(2,\;4,\;6\right) & \left(1,\;2,\;4\right)\\ \left(2,\;4,\;6\right) & \left(3,\;5,\;7\right) & 1 & \left(1,\;3,\;5\right) & \left(3,\;5,\;7\right) & \left(3,\;5,\;7\right)\\ 1/(2,\;4,\;6) & \left(1,\;3,\;5\right) & 1/(1,\;3,\;5) & 1 & \left(3,\;5,\;7\right) & \left(3,\;5,\;7\right)\\ 1/(3,\;5,\;7) & 1/(2,\;4,\;6) & 1/(3,\;5,\;7) & 1/(3,\;5,\;7) & 1 & \left(1,\;3,\;5\right)\\ 1/(3,\;5,\;7) & 1/(1,\;2,\;4) & 1/(3,\;5,\;7) & 1/(3,\;5,\;7) & 1/(1,\;3,\;5) & 1 \end{array}\right]$$
$$\tilde{A}(2)=\left[\begin{array}{cccccc} 1 & \left(1,\;3,\;5\right) & 1/(3,\;5,\;7) & 1/(3,\;5,\;7) & \left(2,\;4,\;6\right) & 1/(1,\;3,\;5)\\ 1/(1,\;3,\;5) & 1 & 1/(3,\;5,\;7) & 1/(3,\;5,\;7) & 1/(1,\;3,\;5) & \left(1,\;2,\;4\right)\\ \left(3,\;5,\;7\right) & \left(3,\;5,\;7\right) & 1 & 1/(1,\;3,\;5) & \left(1,\;3,\;5\right) & \left(3,\;5,\;7\right)\\ \left(3,\;5,\;7\right) & \left(3,\;5,\;7\right) & \left(1,\;3,\;5\right) & 1 & \left(4,\;6,\;8\right) & \left(3,\;5,\;7\right)\\ 1/(2,\;4,\;6) & \left(1,\;3,\;5\right) & 1/(1,\;3,\;5) & 1/(4,\;6,\;8) & 1 & \left(1,\;3,\;5\right)\\ \left(1,\;3,\;5\right) & 1/(1,\;2,\;4) & 1/(3,\;5,\;7) & 1/(3,\;5,\;7) & 1/(1,\;3,\;5) & 1 \end{array}\right]$$
$$\tilde{A}(3)=\left[\begin{array}{cccccc} 1 & 1/(1,\;3,\;5) & 1/(1,\;3,\;5) & 1/(3,\;5,\;7) & 1/(1,\;2,\;4) & \left(1,\;3,\;5\right)\\ \left(1,\;3,\;5\right) & 1 & 1/(2,\;4,\;6) & 1/(1,\;3,\;5) & 1/(3,\;5,\;7) & 1/(1,\;3,\;5)\\ \left(1,\;3,\;5\right) & \left(2,\;4,\;6\right) & 1 & 1/(1,\;2,\;4) & \left(1,\;3,\;5\right) & \left(4,\;6,\;8\right)\\ \left(3,\;5,\;7\right) & \left(1,\;3,\;5\right) & \left(1,\;2,\;4\right) & 1 & \left(2,\;4,\;6\right) & \left(3,\;5,\;7\right)\\ \left(1,\;2,\;4\right) & \left(3,\;5,\;7\right) & 1/(1,\;3,\;5) & 1/(2,\;4,\;6) & 1 & \left(1,\;3,\;5\right)\\ 1/(1,\;3,\;5) & \left(1,\;3,\;5\right) & 1/(4,\;6,\;8) & 1/(3,\;5,\;7) & 1/(1,\;3,\;5) & 1 \end{array}\right]$$

Obviously, there were considerable differences between expert judgments. The consistency ratios of the fuzzy judgment matrixes were evaluated as
$$\tilde{CR}(1)=(-0.691,\;0.216,\;7.225)$$
$$\tilde{CR}(2)=(-0.699,\;0.211,\;8.422)$$
$$\tilde{CR}(3)=(-0.722,\;0.161,\;9.980)$$
which showed a certain degree of consistency since CR ≤ 0.1 ~ 0.3. In addition, the relative priorities of factors evaluated by experts are summarized in Figure 9. Expert #1 considered “the speed of vaccine acquisition” as the most important factor, while the other experts gave higher weights to “demand shrinkage”.

The overall consensus among experts, in terms of the FI results of their evaluations, is shown in Figure 10. Obviously, the FI result was always a nonempty set, showing that an overall consensus existed among all experts.

However, due to the very narrow coverage of the FI result, the overall consensus on the value of ${\tilde{w}}_{2}$, ${\tilde{w}}_{5}$, or ${\tilde{w}}_{6}$ was not enough. In order to solve this problem, the threshold of the FI width was set to 0.1, because the priorities evaluated by experts were not narrower than 0.1. The narrowest priority had a range of 0.093. Then, the partial consensus among most experts was sought instead until the PCFI result became wider than the threshold. In addition, the number of experts was as many as possible. In the end, the 2/3 PCFI result met the two requirements, as shown in Figure 11. ${\tilde{w}}_{4}$ had the maximum value, while ${\tilde{w}}_{6}$ had the minimum value. Therefore, the most influential factor was “the speed of vaccine acquisition”, followed by “demand shrinkage” and “the performance of COVID-19 containment”, while the least influential factor was “infection risk”.

To facilitate the subsequent calculation, the PCFI results were approximated with TFNs according to Equation (18). The approximation results are summarized in Table 3.

Among the six factors, only “pandemic containment performance” was the-higher-the-better performance, whereas the others were the-lower-the-better performances. The performances in optimizing these factors were evaluated according to the rules in Table 4.

The data of the wafer fab in the six aspects are summarized in Table 5, based on which the performances of the wafer fab in optimizing the six factors were evaluated. The evaluation results are summarized in Table 6.

Finally, the robustness of the wafer fab to the COVID-19 pandemic, evaluated using FWA, was (1.500, 3.254, 5.660) with a COG of 3.471. According to the experimental results,
(1)The robustness of the wafer fab was much higher than the neutral value (i.e., 2.5), which indicates that the wafer fab was likely to survive the COVID-19 pandemic.(2)A sensitivity analysis was conducted by increasing the performance in optimizing each factor by 10% (if possible). The improvement in the robustness of the wafer fab was observed. The results are summarized in Table 7. By lessening the shrinkage in demand, the robustness of the wafer fab to the COVID-19 pandemic could be enhanced most. Based on this, the wafer fab could set its action targets.

(1)However, due to the lack of relevant information, the robustness of the wafer fab was not compared with that of another.(2)The robustness of the wafer fab to the COVID-19 pandemic can still be enhanced by taking short-term measures, such as wearing masks, physical distancing, video conferencing, regular disinfection, etc.

## 5. Discussion

Based on the analysis results, the following discussions were conducted:(1)The purpose of proposing the fuzzy collaborative intelligence approach is to assess whether the normal operation of a factory is not affected by the COVID-19 pandemic, not to prevent the outbreak of the COVID-19 pandemic in the factory. However, if the assessment result shows that the factory does not have sufficient robustness to the COVID-19 pandemic, healthcare treatments (such as wearing masks, monitoring body temperature, staggered work shifts, etc.) should be taken to enhance its robustness [77]. It is such healthcare treatments that help prevent the outbreak of the COVID-19 pandemic in the factory.(2)In the experiment, “the speed of vaccine acquisition” and “the performance of COVID-19 containment” were two critical factors to assess the robustness of a factory to the COVID-19 pandemic, which was similar to Tesla’s practice of reopening its factories in China and USA [78]. The two regions have performed well in containing the COVID-19 pandemic and/or developing vaccines [79,80].(3)An existing method, the fuzzy extent analysis (FEA)-FWA method [81], was also applied to the collected data for comparison. In the FEA-FWA method, the relative priority of a critical factor was set to the minimum degree of being the most critical factor, which was a crisp value. Then, the robustness of a factory was assessed using FWA. According to the experimental results, the most critical factor was “the speed of vaccine acquisition”, followed by “demand shrinkage”, which was identical to that drawn using the proposed methodology. The robustness of the wafer fab to the COVID-19 pandemic was evaluated as 3.023, which was much less than that evaluated using the proposed methodology. In our view, the proposed methodology considered the uncertainty of related factors in a more reasonable way. Therefore, the assessment result was more convincing.(4)Before the COVID-19 outbreak, most past studies in this industry focused on assessing and enhancing the competitiveness of a wafer fab [82,83,84,85,86,87,88]. However, after the COVID-19 outbreak, a wafer fab must be robust to survive. Therefore, a wafer fab with low robustness will not have competitiveness.

## 6. Conclusions

The COVID-19 pandemic has affected factories all over the world. However, the impact of the COVID-19 pandemic on different factories is not the same. To investigate this issue, a fuzzy collaborative intelligence approach is proposed in this study to assess the robustness of a factory to the COVID-19 pandemic. The fuzzy collaborative intelligence approach begins with the application of the FGM method to evaluate the relative priorities of factors that affect the robustness of a factory. Then, the evaluation results by all experts are aggregated using FI or PCFI, depending on whether there is an overall consensus among them. Finally, the FWA method is applied to assess the overall performance of the factory; that is, the robustness of the factory to the COVID-19 pandemic. The assessment result can be compared with that of another factory using FTOPSIS. As far as we know, this study is the first attempt at this topic. Most studies in the past have focused on the performance of a local government.

The proposed methodology has been applied to assess the robustness of a wafer fab located in Tainan Scientific Park of Taiwan to illustrate its applicability. A sensitivity analysis was also conducted. After analyzing the experimental results, the following conclusions were drawn:(1)The most critical factor affecting the robustness of the wafer fab was “the speed of vaccine acquisition”, while the least affecting factor was “infection risk” [89].(2)The wafer fab had a fairly strong robustness to the COVID-19 pandemic, showing that it has a good chance to survive the pandemic [45].(3)By reducing the shrinkage in demand for its products, the robustness of the wafer fab could be most effectively enhanced.

When data are available, the robustness of the wafer fab can be compared with that of another wafer fab. To this end, the worksheets for the assessment can be shared in the public domain. In addition, it will be very valuable if all participants in a supply chain perform the same assessment. Further, sometimes an expert dominates the group decision-making process [90], or experts have different levels of authority [91]. These situations should be reflected when designing new methods to enhance the consistency of pairwise comparison results [92]. These constitute some directions for future research.

## Figures and Tables

**Figure 1 healthcare-08-00481-f001:**
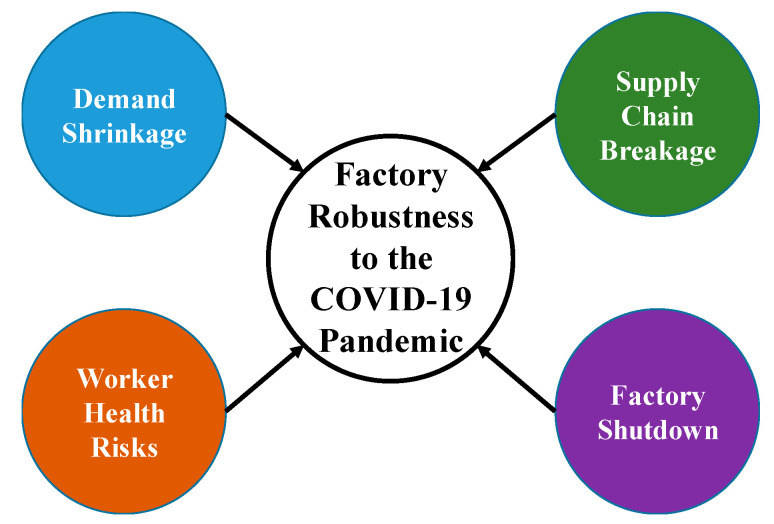
Factors affecting the robustness of a factory to the COVID-19 pandemic.

**Figure 2 healthcare-08-00481-f002:**
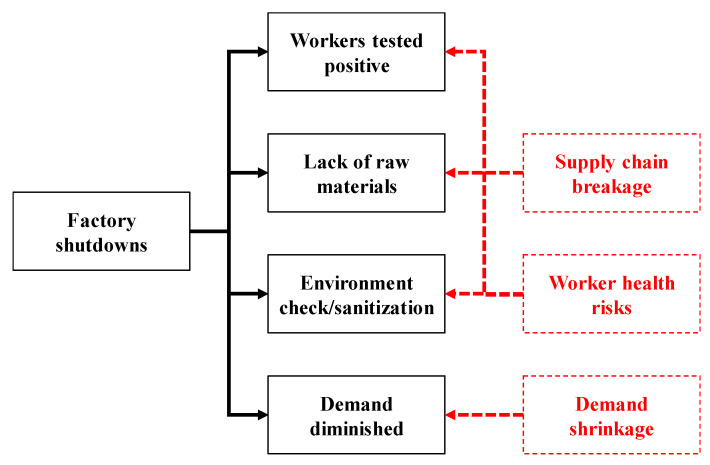
Four types of factory shutdowns amid the COVID-19 pandemic (red parts indicating the causes of shutdowns).

**Figure 3 healthcare-08-00481-f003:**
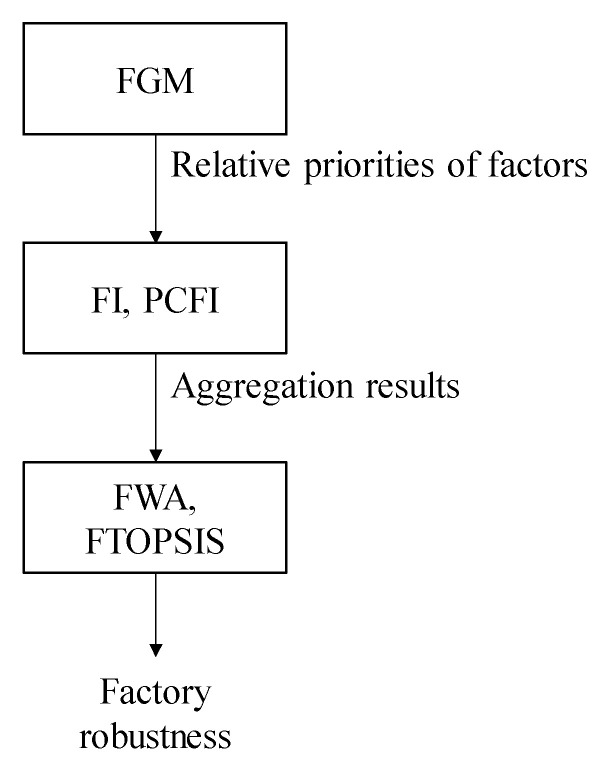
The main parts of the proposed methodology.

**Figure 4 healthcare-08-00481-f004:**
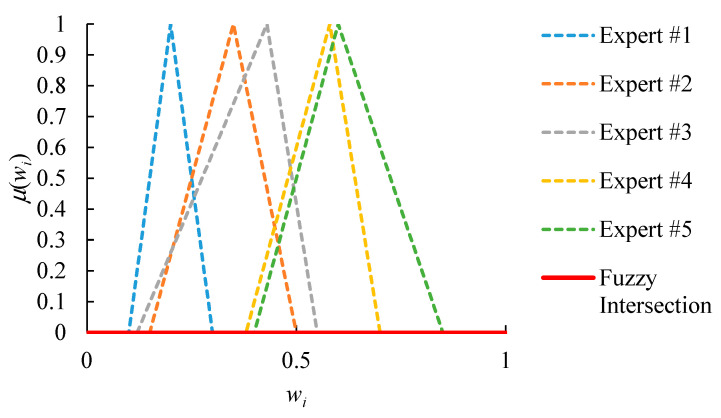
The fuzzy intersection result of five experts’ judgments without a consensus.

**Figure 5 healthcare-08-00481-f005:**
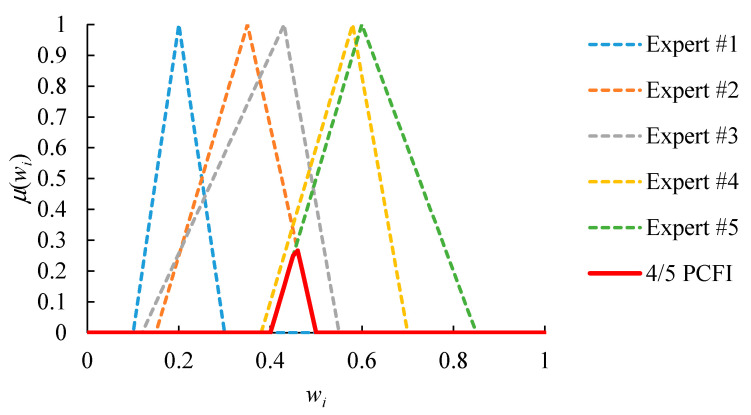
The partial consensus among any four experts in the previous example.

**Figure 6 healthcare-08-00481-f006:**
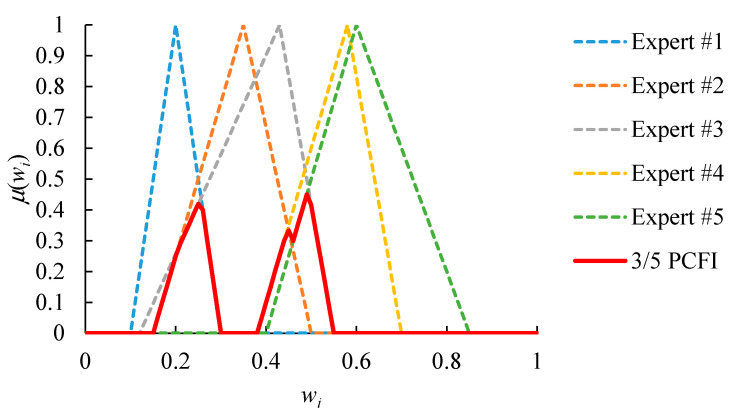
The partial consensus among any three experts in the previous example.

**Figure 7 healthcare-08-00481-f007:**
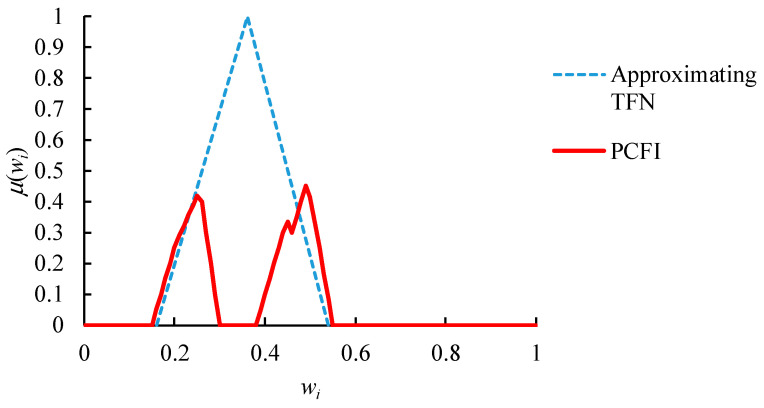
Approximating the PCFI result with a TFN.

**Figure 8 healthcare-08-00481-f008:**
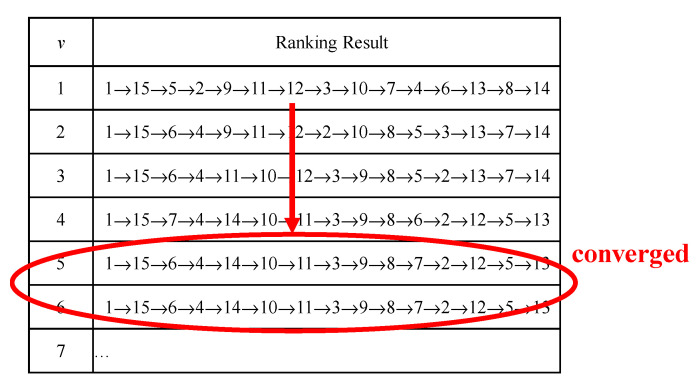
The convergence process of the ranking result.

**Figure 9 healthcare-08-00481-f009:**
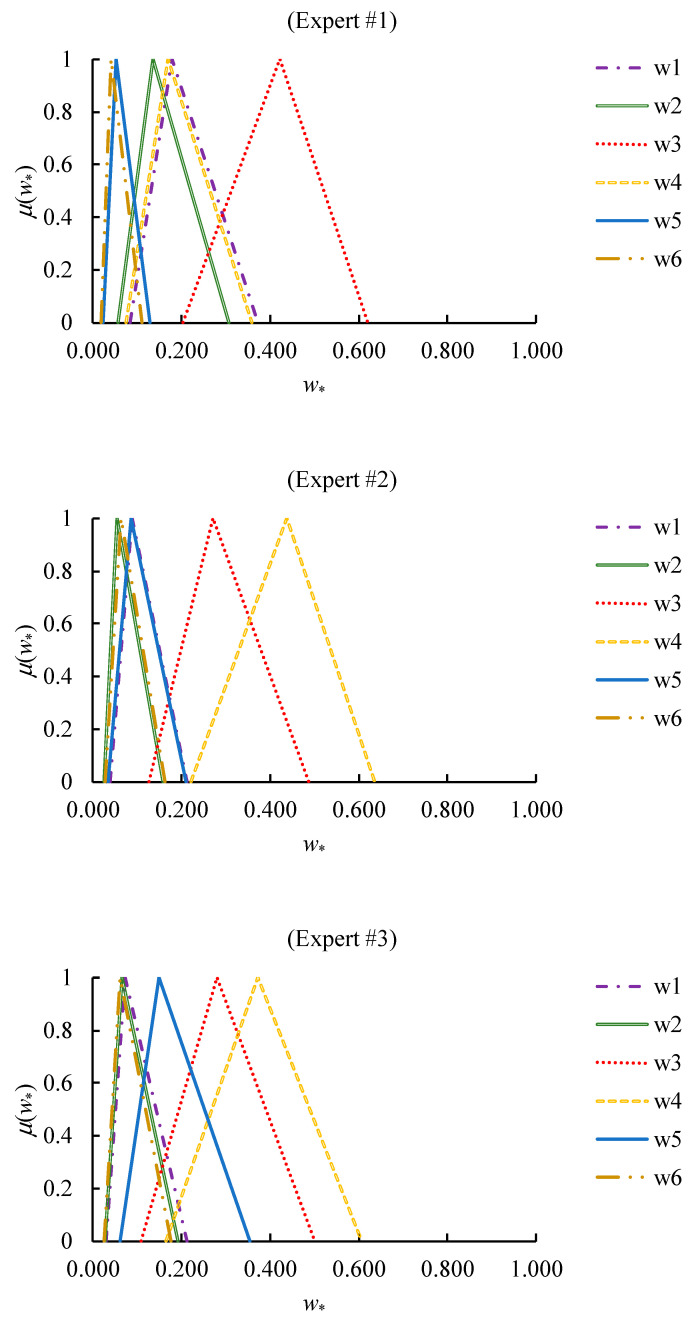
The relative priorities of factors evaluated by experts.

**Figure 10 healthcare-08-00481-f010:**
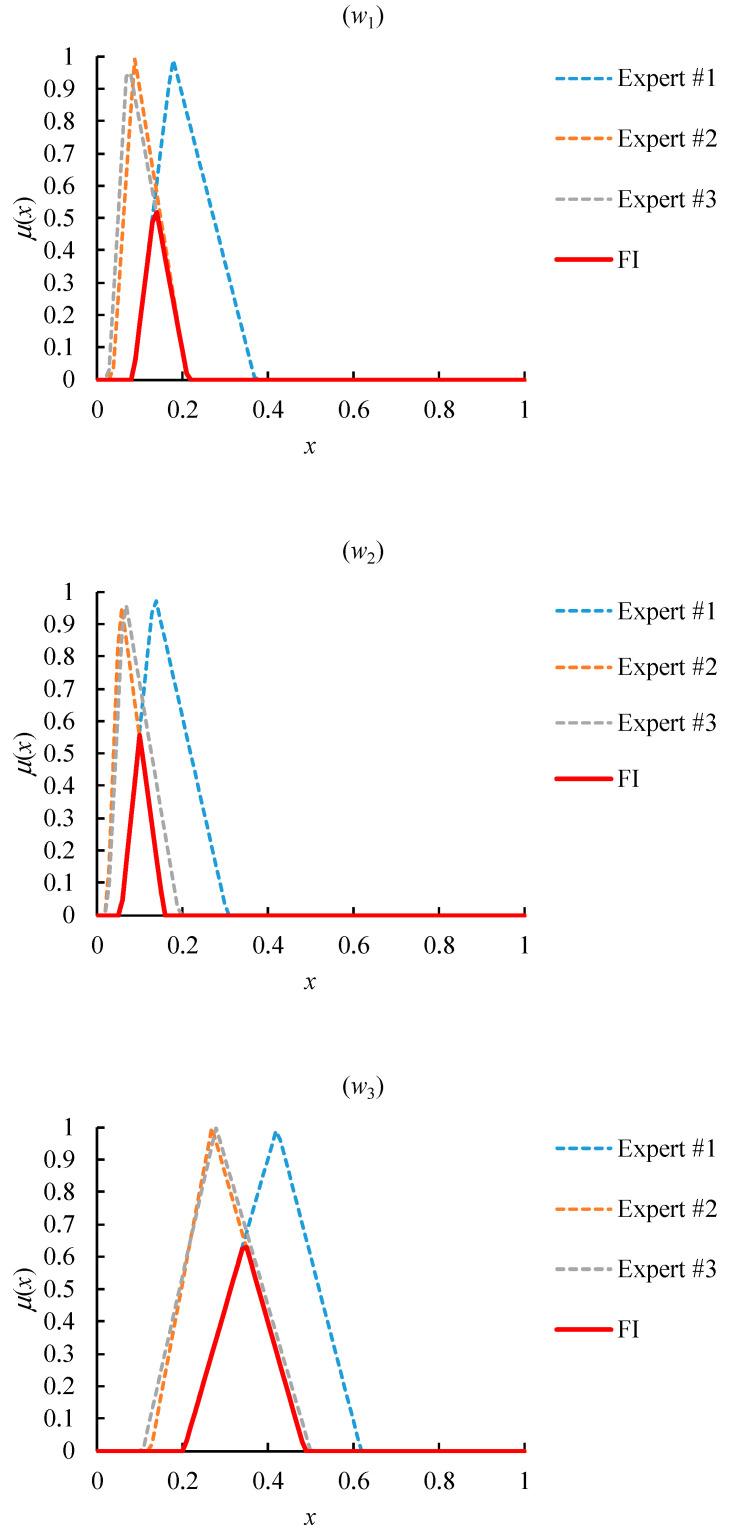
The FI result of the relative priorities evaluated by experts.

**Figure 11 healthcare-08-00481-f011:**
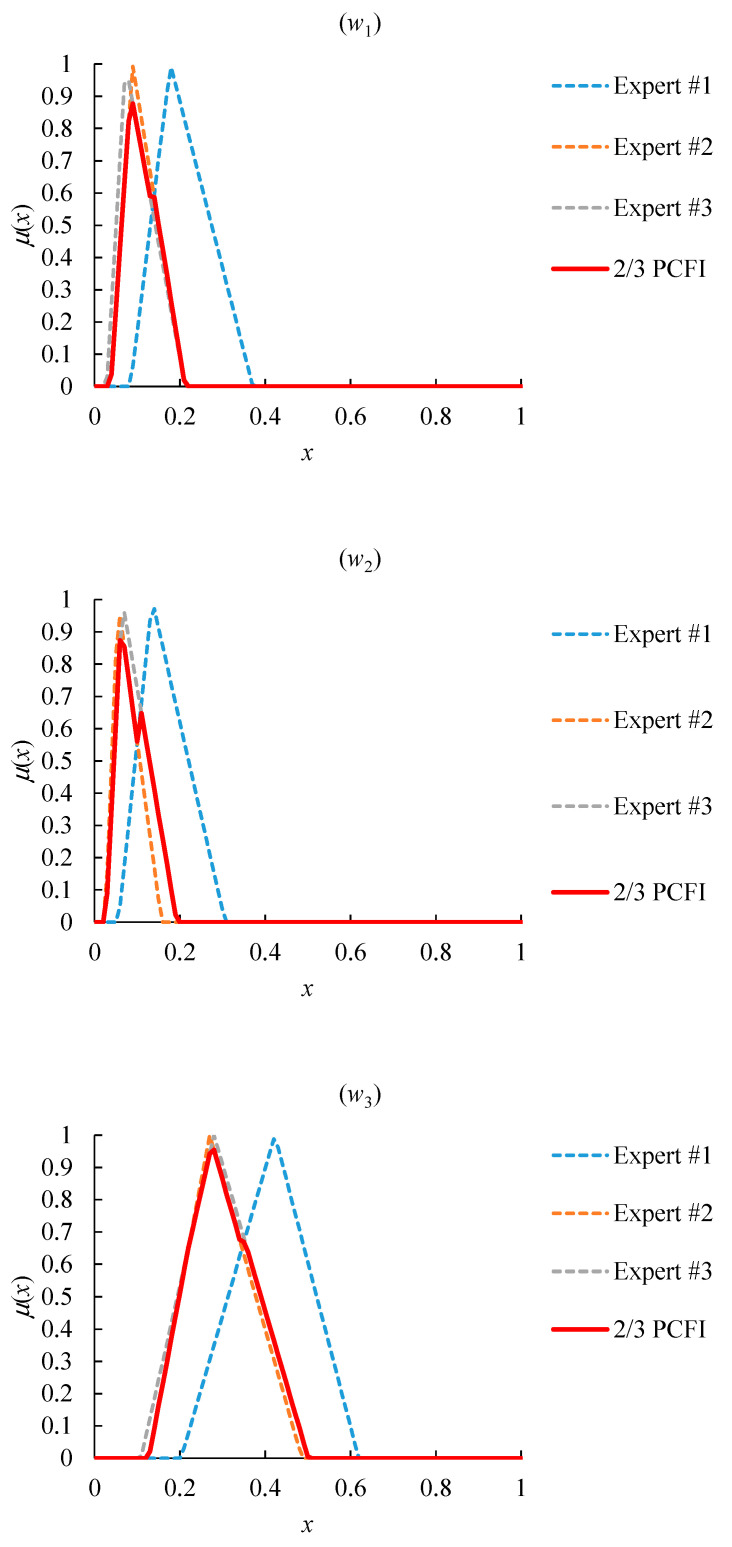
The 2/3 PCFI result.

**Table 1 healthcare-08-00481-t001:** Actions taken by supply chain participants in the short and long terms.

Term	Actions
Short term	Move raw material inventory to places free from quarantine and easy to ship [51] Secure the capacity and delivery plan of upstream raw material suppliers and downstream assembly factories [51] Procure materials that may be in short supply in advance [51] Secure future transportation services [51] Find or activate alternative suppliers [7,51] Negotiate with customers on possible delays or cancellation [51]
Long term	Modify product designs to accept alternative raw materials [51] Establish formal cooperative relations with suppliers outside impacted regions [7,51] Assess the operational and financial risk and impact of the COVID-19 pandemic [51] Modify long-term supply plan and demand forecasts [7,51] Develop factory response/closure plan [7]

**Table 2 healthcare-08-00481-t002:** Results of pairwise comparisons.

**(Expert #1)**		
**Factor #1**	**Factor #2**	**Factor #1 over Factor #2**
Pandemic severity	Pandemic containment performance	Weakly more important than
Vaccine acquisition speed	Pandemic containment performance	Weakly or strongly more important than
Pandemic containment performance	Demand shrinkage	Strongly more important than
Pandemic containment performance	Supplier impact	Strongly more important than
Pandemic containment performance	Infection risk	Strongly more important than
Vaccine acquisition speed	Pandemic severity	Strongly more important than
Demand shrinkage	Pandemic severity	Weakly more important than
Pandemic severity	Supplier impact	Weakly or strongly more important than
Pandemic severity	Infection risk	As equal as or weakly more important than
Vaccine acquisition speed	Demand shrinkage	Weakly more important than
Vaccine acquisition speed	Supplier impact	Strongly more important than
Vaccine acquisition speed	Infection risk	Strongly more important than
Demand shrinkage	Supplier impact	Strongly more important than
Demand shrinkage	Infection risk	Strongly more important than
Supplier impact	Infection risk	Weakly more important than
**(Expert #2)**		
**Factor #1**	**Factor #2**	**Factor #1 over Factor #2**
Pandemic containment performance	Pandemic severity	Weakly more important than
Vaccine acquisition speed	Pandemic containment performance	Strongly more important than
Demand shrinkage	Pandemic containment performance	Strongly more important than
Pandemic containment performance	Supplier impact	Weakly or strongly more important than
Infection risk	Pandemic containment performance	Weakly more important than
Vaccine acquisition speed	Pandemic severity	Strongly more important than
Demand shrinkage	Pandemic severity	Strongly more important than
Supplier impact	Pandemic severity	Weakly more important than
Pandemic severity	Infection risk	As equal as or weakly more important than
Demand shrinkage	Vaccine acquisition speed	Weakly more important than
Vaccine acquisition speed	Supplier impact	Weakly more important than
Vaccine acquisition speed	Infection risk	Strongly more important than
Demand shrinkage	Supplier impact	Strongly or very strongly more important than
Demand shrinkage	Infection risk	Strongly more important than
Supplier impact	Infection risk	Weakly more important than
**(Expert #3)**		
**Factor #1**	**Factor #2**	**Factor #1 over Factor #2**
Pandemic severity	Pandemic containment performance	Weakly more important than
Vaccine acquisition speed	Pandemic containment performance	Weakly more important than
Demand shrinkage	Pandemic containment performance	Strongly more important than
Supplier impact	Pandemic containment performance	As equal as or weakly more important than
Pandemic containment performance	Infection risk	Weakly more important than
Vaccine acquisition speed	Pandemic severity	Weakly or strongly more important than
Demand shrinkage	Pandemic severity	Weakly more important than
Supplier impact	Pandemic severity	Strongly more important than
Infection risk	Pandemic severity	Weakly more important than
Demand shrinkage	Vaccine acquisition speed	As equal as or weakly more important than
Vaccine acquisition speed	Supplier impact	Weakly more important than
Vaccine acquisition speed	Infection risk	Strongly or very strongly more important than
Demand shrinkage	Supplier impact	Weakly or strongly more important than
Demand shrinkage	Infection risk	Strongly more important than
Supplier impact	Infection risk	Weakly more important than

**Table 3 healthcare-08-00481-t003:** Approximating TFNs.

*i*	$\tilde{PCFI}(\{{\tilde{w}}_{i}(m)\}$
1	(0.090, 0.144, 0.210)
2	(0.060, 0.105, 0.150)
3	(0.210, 0.345, 0.480)
4	(0.230, 0.291, 0.350)
5	(0.070, 0.096, 0.120)
6	(0.030, 0.065, 0.110)

**Table 4 healthcare-08-00481-t004:** Rules for evaluating the performances.

Factor	Rule
Pandemic containment performance	${\tilde{p}}_{k1}(x_{k})=\{\begin{array}{ccc} \left(0,\;0,\;1\right) & \mathrm{if} & x_{k}<0.9\cdot \underset{r}{\min}\;x_{r}+0.1\cdot \underset{r}{\max}\;x_{r}\;\mathrm{or}\;\mathrm{data}\;\mathrm{not}\;\mathrm{available}\\ \left(0,\;1,\;2\right) & \mathrm{if} & 0.9\cdot \underset{r}{\min}\;x_{r}+0.1\cdot \underset{r}{\max}\;x_{r}\leq x_{k}<0.65\cdot \underset{r}{\min}\;x_{r}+0.35\cdot \underset{r}{\max}\;x_{r}\\ \left(1.5,\;2.5,\;3.5\right) & \mathrm{if} & 0.65\cdot \underset{r}{\min}\;x_{r}+0.35\cdot \underset{r}{\max}\;x_{r}\leq x_{k}<0.35\cdot \underset{r}{\min}\;x_{r}+0.65\cdot \underset{r}{\max}\;x_{r}\\ \left(3,\;4,\;5\right) & \mathrm{if} & 0.35\cdot \underset{r}{\min}\;x_{r}+0.65\cdot \underset{r}{\max}\;x_{r}\leq x_{k}<0.1\cdot \underset{r}{\min}\;x_{r}+0.9\cdot \underset{r}{\max}\;x_{r}\\ \left(4,\;5,\;5\right) & \mathrm{if} & 0.1\cdot \underset{r}{\min}\;x_{r}+0.9\cdot \underset{r}{\max}\;x_{r}\leq x_{k} \end{array}$ where $x_{k}$ is the recovery index of the region [73].
Pandemic severity	${\tilde{p}}_{k2}(x_{k})=\{\begin{array}{ccc} \left(0,\;0,\;1\right) & \mathrm{if} & 0.1\cdot \underset{r}{\min}\;x_{r}+0.9\cdot \underset{r}{\max}\;x_{r}\leq x_{k}\;\mathrm{or}\;\mathrm{data}\;\mathrm{not}\;\mathrm{available}\\ \left(0,\;1,\;2\right) & \mathrm{if} & 0.35\cdot \underset{r}{\min}\;x_{r}+0.65\cdot \underset{r}{\max}\;x_{r}\leq x_{k}<0.1\cdot \underset{r}{\min}\;x_{r}+0.9\cdot \underset{r}{\max}\;x_{r}\\ \left(1.5,\;2.5,\;3.5\right) & \mathrm{if} & 0.65\cdot \underset{r}{\min}\;x_{r}+0.35\cdot \underset{r}{\max}\;x_{r}\leq x_{k}<0.35\cdot \underset{r}{\min}\;x_{r}+0.65\cdot \underset{r}{\max}\;x_{r}\\ \left(3,\;4,\;5\right) & \mathrm{if} & 0.9\cdot \underset{r}{\min}\;x_{r}+0.1\cdot \underset{r}{\max}\;x_{r}\leq x_{k}<0.65\cdot \underset{r}{\min}\;x_{r}+0.35\cdot \underset{r}{\max}\;x_{r}\\ \left(4,\;5,\;5\right) & \mathrm{if} & x_{k}<0.9\cdot \underset{r}{\min}\;x_{r}+0.1\cdot \underset{r}{\max}\;x_{r} \end{array}$ where $x_{k}$ is the current number of active cases in the region.
Vaccine acquiring speed	${\tilde{p}}_{k3}(x_{k})=\{\begin{array}{ccc} \left(0,\;0,\;1\right) & \mathrm{if} & x_{k}=\mathrm{no}\;\mathrm{vaccine}\;\mathrm{or}\;\mathrm{data}\;\mathrm{not}\;\mathrm{available}\\ \left(0,\;1,\;2\right) & \mathrm{if} & x_{k}=\mathrm{phase}\;1\\ \left(1.5,\;2.5,\;3.5\right) & \mathrm{if} & x_{k}=\mathrm{phase}\;2\\ \left(3,\;4,\;5\right) & \mathrm{if} & x_{k}=\mathrm{phase}\;3\\ \left(4,\;5,\;5\right) & \mathrm{if} & x_{k}=\mathrm{approved}\;\mathrm{for}\;\mathrm{limited}\;\mathrm{or}\;\mathrm{full}\;\mathrm{use} \end{array}$ where $x_{k}$ is the stage of vaccines developed in the region [74].
Demand shrinkage	${\tilde{p}}_{k4}(x_{k})=\{\begin{array}{ccc} \left(0,\;0,\;1\right) & \mathrm{if} & x_{k}=\mathrm{very}\;\mathrm{signficant}\;\mathrm{or}\;\mathrm{data}\;\mathrm{not}\;\mathrm{available}\\ \left(0,\;1,\;2\right) & \mathrm{if} & x_{k}=\mathrm{significant}\\ \left(1.5,\;2.5,\;3.5\right) & \mathrm{if} & x_{k}=\mathrm{moderate}\\ \left(3,\;4,\;5\right) & \mathrm{if} & x_{k}=\mathrm{insignificant}\\ \left(4,\;5,\;5\right) & \mathrm{if} & x_{k}=\mathrm{very}\;\mathrm{insignificant}\;\mathrm{or}\;\mathrm{demand}\;\mathrm{increased} \end{array}$ where $x_{k}$ is the shrinkage of demand.
Supplier impact	${\tilde{p}}_{k5}(x_{k})=\{\begin{array}{ccc} \left(0,\;0,\;1\right) & \mathrm{if} & x_{k}=\mathrm{very}\;\mathrm{high}\;\mathrm{or}\;\mathrm{data}\;\mathrm{not}\;\mathrm{availeble}\\ \left(0,\;1,\;2\right) & \mathrm{if} & x_{k}=\mathrm{high}\\ \left(1.5,\;2.5,\;3.5\right) & \mathrm{if} & x_{k}=\mathrm{moderate}\\ \left(3,\;4,\;5\right) & \mathrm{if} & x_{k}=\mathrm{low}\\ \left(4,\;5,\;5\right) & \mathrm{if} & x_{k}=\mathrm{very}\;\mathrm{low} \end{array}$ where $x_{k}$ is the impact on suppliers.
Infection risk	${\tilde{p}}_{k6}(x_{k})=\{\begin{array}{ccc} \left(0,\;0,\;1\right) & \mathrm{if} & x_{k}=\mathrm{very}\;\mathrm{high}\;\mathrm{or}\;\mathrm{data}\;\mathrm{not}\;\mathrm{available}\\ \left(0,\;1,\;2\right) & \mathrm{if} & x_{k}=\mathrm{high}\\ \left(1.5,\;2.5,\;3.5\right) & \mathrm{if} & x_{k}=\mathrm{moderate}\\ \left(3,\;4,\;5\right) & \mathrm{if} & x_{k}=\mathrm{low}\\ \left(4,\;5,\;5\right) & \mathrm{if} & x_{k}=\mathrm{very}\;\mathrm{low} \end{array}$ where $x_{k}$ is the risk of infection.

**Table 5 healthcare-08-00481-t005:** Data of the wafer fab in the six aspects.

Factor	Data (Collected on October 2)
Pandemic containment performance	The recovery index of the region was 77.99 [75]. The maximum and minimum of the recovery index were 87.24 (Australia) and 9.38 (Honduras).
Pandemic severity	The current number of active cases in the region was 24 [76]. The maximum and minimum of the current number of active cases were 2,545,390 (USA) and 1 (Mongolia and four other countries).
Vaccine acquisition speed	Three locally developed COVID-19 vaccines were in phase 1.
Demand shrinkage	The demand for major products, integrated circuits for wireless networks and panel drivers, did not shrink significantly.
Supplier impact	Supplier were located in Taiwan (very lowly impacted), Netherlands (highly impacted), and USA (very highly impacted).
Infection risk	Travelling abroad for business and personal trips has been suspended and discouraged. All types of wafer fabrication equipment are operated in a clean room. Employees wear clean room suits at all times. The probability of infection within the wafer fab is small. Most types of wafer fabrication equipment are highly automated. Employees work independently. There is little chance of contact with each other.

**Table 6 healthcare-08-00481-t006:** Evaluation results.

*i*	${\tilde{p}}_{i}$
1	(3, 4, 5)
2	(4, 5, 5)
3	(0, 1, 2)
4	(3, 4, 5)
5	(3, 4, 5)
6	(3, 4, 5)

**Table 7 healthcare-08-00481-t007:** Robustness of the wafer fab after improving the performance in optimizing each factor.

*i*	$COG(\tilde{O})$
1	3.503
2	3.471
3	3.506
4	3.541
5	3.493
6	3.484

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
