# Peer review of "Assessing the Robustness of a Factory Amid the COVID-19 Pandemic: A Fuzzy Collaborative Intelligence Approach"

_healthcare, 2020, doi:10.3390/healthcare8040481_

Round 1

Reviewer 1 Report

The authors assessed the robustness of a factory based on fuzzy collaborative intelligence. However, this article is rather a case study than a research paper. That is why I do not recommend the article for publishing in the present form. Below I listed some detailed comments showing that the article should have more attention during the writing.

  1. Although the authors present the impact of the COVID-19 pandemic on factories, the necessity to assess the robustness of a factory is not enough. Give more presentation in Introduction.
  2. What are the disadvantages of ordinary assessment techniques? Why do you choose fuzzy collaborative intelligence rather than others?
  3. I don't see the text about Discussion. What is the relationship between your research findings and other scholars? Do you support or oppose their opinions? What's your contribution to the prevention of COVID-19 pandemic?
  4. About one half of the references are websites; it seems that this paper is lowly technicality.

Reviewer 2 Report

The manuscript "Assesing the robustness of a factory amid the COVID-19 pandemic: A fuzzy collaborative intelligent approach" describes an interesting strategy to evaluate priorities of diverse factors having an effect on the robustness of a factory. COVID-19 pandemic has turned factories vulnerable to diverse factors such product and demand shinkkage, helath risks, supply chain breakage, and factory shutdown. A fuzzy collaborative intelligence system taking into account multiple expert experience is used to determine robustness.

The proposed method is properly described, and the application is quite interesting and relevant nowdays. 

Some minor comments:

Section 2 Literature reviews. This section seems to me more a background on factors having effect on productivity rather than a literature review, so I suggest to rename this section

Avoid phrases such "As a result, some factories are robust to COVID-19 pandemic, while others are not", this is evident!

Reference 18, delete the symbol "&" among the authors.

Reviewer 3 Report

The paper proposes a fuzzy collaborative intelligence approach to assess the robustness of a factory amid the COVID-19 pandemic, in which fuzzy AHP and fuzzy TOPSIS/fuzzy weighted average approach are applied. Overall, this is an interesting application paper. My detailed comments can be found in the attachment.

Reviewer 4 Report

A fuzzy collaborative intelligence approach is proposed  in order to evaluate  the
robustness of a factory to the COVID-19 pandemic.

The authors define the term robustness of a factory to the COVID-19 pandemic with "the degree to which the normal operation of a factory is not affected by the COVID 19 pandemic, without taking short-term preventive measures". in the introduction, to motivate the use of a fuzzy collaborative intelligence approach to evaluate the impacts of the COVID-19 pandemic, highlight that this concept of robustness is a fuzzy concept rather than a random variable manageable with probability theory. is a factory robust or resilient to the COVID-19 pandemic certainly also depends on the hazard scenario against which the impacts / risks are measured, or on the local spread scenario of the pandemic that is considered. This should be specified more accurately in the introductory section.

Authors must accurately specify the meaning of all concepts and symbols in the equations. For example, in line 153 the aij components of the fuzzy judgment matrix are reported with a tilde symbol, but the meaning of this symbol is not specified. On line 159 the fuzzy subtraction and multiplication operators are not defined and specified. Similarly, the concept of fuzzy maximal eigenvalue is not specified in line 166 and it is necessary to define, taking it from the well-known literature, the concept of consistency on p. 172.

Specific care must be taken in defining all the symbols in equations 16,17,18, 19 and 25. Furthermore, equation 18 must be structured correctly.

A description in structured mode, for example through the use of a pseudocode, is necessary to show and list the steps of the proposed method.

On line 326 it is necessary that the authors justify the choice of the threshold for the width of the FI result to 0.1.

The experimentation carried out lacks comparative tests of the results with that of other methods used for the evaluation of the robustness to impacts produced by disease spreading. Authors need to perform these tests to assess the accuracy of the proposed method.

Round 2

Reviewer 1 Report

Well done!

Reviewer 3 Report

The paper has been improved and is acceptable.

Reviewer 4 Report

In this new version of their paper the authors take into account all my suggestions. I consider this paper publishable in the current form.